# Copy Number Variant Detection with Low-Coverage Whole-Genome Sequencing Represents a Viable Alternative to the Conventional Array-CGH

**DOI:** 10.3390/diagnostics11040708

**Published:** 2021-04-15

**Authors:** Marcel Kucharík, Jaroslav Budiš, Michaela Hýblová, Gabriel Minárik, Tomáš Szemes

**Affiliations:** 1Geneton s.r.o., 841 04 Bratislava, Slovakia; jaroslav.budis@geneton.sk (J.B.); tomas.szemes@geneton.sk (T.S.); 2Comenius University Science Park, 841 04 Bratislava, Slovakia; 3Slovak Centre of Scientific and Technical Information, 811 04 Bratislava, Slovakia; 4Trisomy Test s.r.o., 841 04 Bratislava, Slovakia; michaela.hyblova@medirex.sk (M.H.); gabriel.minarik@medirex.sk (G.M.); 5Department of Molecular Biology, Faculty of Natural Sciences, Comenius University, 842 15 Bratislava, Slovakia

**Keywords:** CNV detection, low-coverage WGS, CNV detection comparison, aCGH replacement

## Abstract

Copy number variations (CNVs) represent a type of structural variant involving alterations in the number of copies of specific regions of DNA that can either be deleted or duplicated. CNVs contribute substantially to normal population variability, however, abnormal CNVs cause numerous genetic disorders. At present, several methods for CNV detection are applied, ranging from the conventional cytogenetic analysis, through microarray-based methods (aCGH), to next-generation sequencing (NGS). In this paper, we present GenomeScreen, an NGS-based CNV detection method for low-coverage, whole-genome sequencing. We determined the theoretical limits of its accuracy and obtained confirmation in an extensive in silico study and in real patient samples with known genotypes. In theory, at least 6 M uniquely mapped reads are required to detect a CNV with the length of 100 kilobases (kb) or more with high confidence (Z-score > 7). In practice, the in silico analysis required at least 8 M to obtain >99% accuracy (for 100 kb deviations). We compared GenomeScreen with one of the currently used aCGH methods in diagnostic laboratories, which has mean resolution of 200 kb. GenomeScreen and aCGH both detected 59 deviations, while GenomeScreen furthermore detected 134 other (usually) smaller variations. When compared to aCGH, overall performance of the proposed GenemoScreen tool is comparable or superior in terms of accuracy, turn-around time, and cost-effectiveness, thus providing reasonable benefits, particularly in a prenatal diagnosis setting.

## 1. Introduction

Copy number variations (CNVs) represent a phenomenon in which sections of the genome are repeated while the number of repeats in the genome varies between individuals. CNVs contribute substantially to normal population variability. However, abnormal CNVs are known to cause numerous genetic disorders. Several methods for CNV analysis are used, from the conventional cytogenetic analysis, through microarray-based approaches, to next-generation sequencing (NGS) [1].

Array-based comparative genomic hybridization (aCGH) delivers genome-wide coverage at a great resolution, even on the scale of dozens of kilobases (10–25 kb) [2]. This fact resulted in aCGH having been the gold standard in CNVs detection for several years. Even though current microarrays offer flexibility in coverage across variable resolution formats, there are still some disadvantages to be considered. For example, in prenatal diagnosis from amniotic fluid, micrograms of genomic DNA are typically needed to hybridize to an array. This can be accomplished either by time-consuming culturing taking up to two weeks, or by whole-genome amplification, which can introduce bias into the analysis. On the contrary, NGS utilizes mere nanograms of DNA, thus not requiring additional amplification. There is lower likelihood of sample contamination due to less material required. The transition from the proven microarray platform to NGS often reveals some new and unexpected data; however, it seems to be a very slow event though the cost and time aspect is already quite unprecedented. Additionally, while aCGH equipment serves a single purpose only, commonly used NGS platforms are very versatile, enabling numerous applications, including exome, genome, targeted panels, transcriptome, or episome sequencing. The whole-exome and targeted sequencing aims to reduce the sequencing cost but is limited to certain regions (protein-coding or custom), where most known disease-causing mutations occur [3]. NGS provides a sensitive and accurate approach for the detection of the major types of genomic variations, including CNVs [4,5].

A handful of CNV detection tools have been introduced in recent years, specifically for targeted and exome sequencing [6,7,8,9,10,11,12]. However, these tools are not suitable for data from whole-genome, low-coverage sequencing. The notable whole-genome CNV detection tools include Wisecondor X [13] (successor of Wisecondor [14] tool), CNVkit [15], CNVnator [16], or iCopyDav [17]. Partial comparison of some of these tools is provided in the publication of Wisecondor X [13].

In this paper, we present GenomeScreen—a low-coverage, whole-genome NGS-based CNV detection method and estimate its accuracy in theoretical and in silico settings. This method is partially based on the previously published non-invasive prenatal testing (NIPT) CNV detection method [18,19]. The main differences are the parameters of the reported CNVs—in the NIPT setting, the CNVs corresponding to more than 5% fetal fraction and at least 3 Mb in size were reported. Here, on the other hand, we focus on full (non-mosaic) aberrations with much shorter length (100 kb and larger). Furthermore, we compare the sensitivity of GenomeScreen to the more conventional aCGH method on 106 laboratory-prepared clinical samples. The comparison of GenomeScreen and different CNV detection tools goes beyond the scope of this article due to focus on the comparison with the aCGH method itself.

## 2. Materials and Methods

### 2.1. Sample Collection and Processing

All patient samples were analyzed as a part of commercially available testing in cooperation with gynecologists, clinical geneticists, and genetic centers. All patients signed informed consent regarding participation in the research project. Samples of chorionic villi, amniotic fluid, placenta, tissue, or peripheral blood were obtained from 106 patients in the clinical sample group and 789 patients in the training group. Peripheral blood was sampled in K2E (EDTA) vacuum tubes (BD Vacutainer, Plymouth, UK) or Cell-Free DNA BCT (STRECK) vacuum tubes (Streck, La Vista, NE, USA), inverted several times after collection, stored in chilled environment (4–10 °C) for EDTA and at room temperature for STRECK tubes, and transported to the laboratory within 36 h. DNA was extracted from 200 µL of whole blood or 700 µL of amniotic fluid using the QIAamp DNA Blood Mini Kit (Qiagen, Hilden, Germany) according to the manufacturer’s protocol and stored at −20 °C until further analysis.

Genomic DNA from clinical samples was fragmented using 1 U/μL dsDNA Shearase™ Plus (Zymo Research, Irvine, CA, USA) and incubated for 23 min at 42 °C to generate 100–500 bp fragments. For adapter-ligated DNA library construction, the TruSeq Nano kit (Illumina, San Diego, CA, USA) with an in-house optimized protocol was used. Low-coverage sequencing (0.3×) was performed on the Illumina NextSeq 500/550 platform (Illumina, San Diego, CA, USA) with paired-end setting 2 × 35 using High-Output Sequencing Kit v2.5. Library quantity and quality were measured by fluorometric assay on Qubit 2.0 (dsDNA HS Assay Kit, Life Technologies, Eugene, OR, USA). Fragment analysis was performed on the 2100 Bioanalyzer (High Sensitivity DNA Kit, Agilent Technologies, Waldbronn, Germany). We targeted 5 M uniquely mapped reads per sample, while none of the analyses were excluded due to lower (or higher) read counts (more details in Appendix A).

### 2.2. Theoretical Minimal Read Count Estimation

Let us suppose that we model sequencing as a random choice of reads from the whole (mappable) genome. Then, we can theoretically deduce the number of necessary uniquely mapped reads for a certain accuracy criterion. The random choice for a target region is described by the binomial distribution with the mean μ=np and the variance $\sigma^{2}=np\left(1-p\right)$. Here, p is the probability of choosing a read from the target region, and n is the number of reads sequenced. The probability p can furthermore be expressed as the ratio of the region length $l_{c}$ to the whole-genome length $l_{g}$ ($p=l_{c}/l_{g}$). When predicting a CNV, we need to have a certain confidence traditionally determined by the Z-score (Z), defined as follows:(1)$$Z=\frac{\delta -\mu }{\sigma }$$

Here, δ represents the number of reads that we observe in the target region. We assume that the number of reads in the target region will be proportional to the number of present copies of gonosomes, i.e., either δ=n(p+p/2) for duplication or δ=n(p−p/2) for deletion of the region on a single chromosome. If we solve the equation for $Z^{2}$ and substitute
(2)$$Z^{2}=\frac{{\left(\delta -\mu \right)}^{2}}{\sigma^{2}}=\frac{{\left(n\left(p+p/2\right)-np\right)}^{2}}{np\left(1-p\right)}=\frac{n^{2}p^{2}}{4np\left(1-p\right)}=\frac{np}{4\left(1-p\right)}=\frac{nl_{c}}{4\left(l_{g}-l_{c}\right)}$$
then we can estimate the minimal number of reads (n) to be able to predict a variation with length $l_{c}$ with the desired Z-score (Z):(3)$$n\geq \frac{4Z^{2}\left(l_{g}-l_{c}\right)}{l_{c}}$$

### 2.3. Variant Identification

To identify variations, we performed the following pipeline:
Mapping and binningMapping reads using Bowtie 2 [20];Binning reads into same-size 20 kb bins;Normalizing bin counts.Normalization (similar to the one published previously by [21])GC bias correction by LOESS smoothing method [22];Principal component analysis (PCA) normalization to remove higher-order population artifacts on autosomal chromosomes;Subtracting per-bin mean bin count to obtain data normalized around zero.Filtration of unusable binsUnmappable or poorly mappable regions (zero or low mean of bin count);Repetitive regions or areas with certain systematically increased mappability (high mean of bin count);Highly variable regions (high variance of bin count).Segment identification and reportingCircular binary segmentation algorithm [23] to identify consistent segments of similar coverage;Assigning significance to segments based on the proportion of reads;Visualization of findings (Figure 1).

Scripts (Python 3.7) and data are available on the website https://github.com/marcelTBI/GenomeScreen (accessed on 14 April 2021).

#### 2.3.1. Mapping and Binning

Firstly, the reads were mapped to a reference using Bowtie 2 [20] with --very-sensitive settings. We used the hg19/GRCh37 reference in all applications, but other references can be used without changes to the algorithm. The reads were then filtered for map quality of at least 40 and binned according to their starts to same-size 20 kb bins. All subsequent analyses were performed on the bin counts, while the algorithm did not use any other information about reads (for example, sequence). For training purposes, the bin counts corresponding to autosomal chromosomes for each sample were normalized to the identical number of reads (i.e., each bin was divided so the sum of all bins on autosomal chromosomes would be the same for each sample). Furthermore, the same was performed separately for chromosome X and chromosome Y. As a consequence of the separate normalization of sex chromosomes, the applied approach can only detect small sex chromosomal variations and not the whole sex chromosomal aneuploidies.

#### 2.3.2. Normalization

Normalization consisted of three steps: firstly, a sample-wise LOESS-based GC correction was deployed on the bin counts [22]. Next, the principal component analysis (PCA) normalization was used to remove higher-order population artifacts on autosomal chromosomes [21]. For training of the PCA, LOESS-corrected bin counts of 789 NIPT samples with female fetuses were converted to principal component space and the first 15 principal components were stored. The bin count vector of a new sample was then transformed into principal component space defined by these first 15 components and transformed back to the bin space to obtain residuals that were then removed from the bin counts. The first principal components represent the noise commonly observed in euploid samples, and their removal facilitates data normalization. In the present case, the PCA normalization was performed only on autosomal chromosomes due to unavailability of a sufficient number of male samples for training. In the future, the training of PCA on both male and female samples is likely to increase the precision of prediction for sex chromosomes. Lastly, we subtracted per-bin mean bin counts to obtain data normalized around zero. This last step was trained already on the PCA normalized bin counts (where available) and helped compensate for the mapping inequality between various genomic regions.

#### 2.3.3. Filtration of Unusable Bins

To further improve accuracy, we filtered bins that had an unusual signature—low mean (this signaled poor mappability of the region), high mean (repetitive regions or regions with a certain systematic bias), or high variance (highly variable regions). Furthermore, the filtered regions were manually curated to reduce their scatter, mainly around centromeres and in sex chromosomes. The filtration screened out around 15% of the genome, mainly due to the low mappability, especially in and around centromeres.

#### 2.3.4. Segment Identification and Reporting

After normalization and filtering, we received a signal (grey dots in Figure 1) that required segmentation into identical -level parts to be evaluated. To this end, we used the circular binary segmentation (CBS) algorithm implemented in the R package DNAcopy [23]. After segmentation, each segment was assigned a significance level based on its length and difference from zero. Since we knew the mean bin counts, we could estimate the level for a complete deletion or duplication per single copy of a chromosome (magenta dashed lines in Figure 1). We then differed between five color-coded levels of significance: magenta—minimum 75%, minimum 200 kb, red—minimum 25%, minimum 200 kb, orange—minimum 25%, minimum 40 kb, yellow—minimum 12.5%, minimum 40 kb, and green—all others (very short segments or segments around zero). The findings were then reported as a text file for further machine processing, while each chromosome was visualized (Figure 1).

### 2.4. In-Silico Analysis

For the in silico analysis, we chose 83 samples without any aberration and with a read count of at least 10 M. Firstly, the samples were down-sampled to the studied read count (3–10 M with the step of 1 M). Then, for each of the tested variation lengths (20–200 kb with the step of 20 kb), 100 random variations on autosomal chromosomes were generated that did not overlap with the filtered regions (see Section 2.3.3). To create a sample with an artificial aberration, the bins corresponding to the generated random variation were multiplied accordingly (thus, the most time-consuming mapping step was performed only once per sample). Next, variant identification was performed without changes.

In total, we gradually created 664,000 artificial samples (100 variations × 83 samples × 10 variation lengths × 8 read counts) and performed variant identification on them to analyze the impact of read count and variant length. Every detection that overlapped the simulated region (the exact match of the coordinates was not required) was reported as successful. 

## 3. Results

### 3.1. Theoretical Minimal Read Count

The theoretical minimum of reads for predicting a variation with length $l_{c}$ with the desired Z-score (Z) is estimated as (see Section 2.2)
(4)$$n\geq \frac{4Z^{2}\left(l_{g}-l_{c}\right)}{l_{c}}$$

As a standard, the Z-score of 4 is used in the detection of whole chromosomal aneuploidies [24,25]; however, there are inherently more possible CNVs than whole chromosomal aneuploidies. Thus, the desired Z-score should be much higher in this instance to reduce the number of false positives. Moreover, in practice, the number of necessary reads would be even higher due to the uncertainty of sequencing and mapping, and inherent biological biases [26,27]. The theoretical minimal read count estimation for different Z-scores is displayed in Figure 2.

### 3.2. Detection Accuracy for Variable CNV Lengths and Read Count (In Silico)

To verify the theoretically estimated limitations, we first conducted a simulated in silico experiment. Artificial samples with simulated CNV were created from healthy samples by multiplication of bins corresponding to the simulated regions randomly selected on the genome. Only the regions that did not span into filtered positions were kept for further analysis (about 85% of the genome). The details can be found in Section 2.3.

The in silico analysis shows the influence of read count and CNV length on prediction accuracy (Figure 3). Based on the findings, we recommend using read counts of at least 8 M to achieve >99% prediction accuracy for variations with 100 kb and more. We therefore recommend following the line for the Z-score of 8 (red on Figure 2) to get an estimation for different CNV lengths.

Comparison of simulated and reported regions showed that the method can predict the exact simulated region coordinates in 88.2% or coordinates with one-bin difference in 97.7% of cases for 200 kb variation length and 10 M reads. These numbers slightly drop to 75.3% and 91.7% for 5 M reads. The imperfection in predicting coordinates is caused by low coverage and lower mappability of some genomic regions.

### 3.3. Validation of Clinical Samples

Finally, we ran an evaluation of samples analyzed previously in diagnostic settings using the aCGH method (Human Genome CGH Microarray 4 × 44 K Agilent [28]) and GenomeScreen. The selected aCGH method has 42,494 probes, which result in mean accuracy of detection of approximately 200 kb. However, the probes are focused mainly in gene regions and very sparsely in intergenomic regions; therefore, accuracy will be higher within the gene regions and lower outside the genes.

From the 106 tested samples, 58 did not show any detection on aCGH, and the rest contained 59 detections in total (lengths from 39 kb to 146 Mb), all of which were also detected by GenomeScreen. The detections on GenomeScreen and on aCGH show excellent concordance—median overlap of 94.37% (more data in Appendix A). GenomeScreen furthermore detected 134 additional variations with ranges from 80 kb to 1.48 Mb, mainly in the regions with a low number of aCGH probes and protein-coding genes, where aCGH has low coverage (Figure 4 and Appendix A).

## 4. Discussion

GenomeScreen test is a result of evolving laboratory methods and bioinformatic tools validated in our laboratory and is currently available commercially. The assay originated from a basic NIPT test focused on noninvasive prenatal screening for the three most common trisomies. Later, the development continued by adding the detection of sex chromosome aneuploidies and five selected microdeletions, and most recently it has been advanced to a whole-genome scan for chromosomal microaberrations [18,24,25]. The common link between all these tests is the method based on low-coverage, whole-genome sequencing. Because all the versions of the above-mentioned NIPT tests are intended only for screening, we wanted to validate the method also for diagnostic purposes with much broader applicability in prenatal and postnatal diagnostics. One of the key applications is the replacement of aCGH as the confirmatory method in noninvasive prenatal diagnostics. Therefore, in the pilot phase, the method was validated on plasma and amniotic fluid samples, while the analysis was later extended to chorionic villi, placental tissue, blood, buffy coat, and fetal tissue.

GenomeScreen uses a binning approach, and the genomic coordinates of detected variations are reported as a multiplier of the bin size (20 kb). Nevertheless, the prediction of exact coordinates of the variation is not perfect (see Section 3.2), and it is therefore not suitable for precise CNV detection at the level of exons. On the other hand, the aCGH method uses probes, which can be seen as variable-size bins, where the resolution is equal to the probe distance (which is sometimes larger than the 20 kb bin size). The precision of both GenomeScreen and aCGH can be easily increased (by decreasing the bin size and deeper sequencing in the case of GenomeScreen, or by introducing new probes in case of aCGH), but these adjustments inevitably bring higher production cost.

The overall accuracy strongly depends on the depth of sequencing (see Figure 3). If we set the GenomeScreen sequencing depth to achieve a slightly higher accuracy compared with aCGH, the cost per sample is 2–3 times lower for GenomeScreen. Furthermore, the turnaround time from submission of a sample to completion of the whole process including the analysis takes less time in the case of GenomeScreen (typically 2 to 5 days), whereas the aCGH process may take up to 2 weeks when culturing is required. The culturing or DNA amplification is usually required in NIPT setting, since the amount of retrieved DNA is not sufficient for direct application of aCGH. However, even without these prior preparations, the hybridization process itself takes at least 3 days to deliver the result.

The disadvantage of GenomeScreen is the necessity to train the used normalization on at least 100 nonaberrated samples (training on fewer samples results in filtration of an unnecessary large number of bins due to high variability), but we recommend using as many samples as possible for training. The training should be performed separately for each sample type (and/or different laboratory protocol), however, the trained parameters are quite close across the different sample types that we studied. The parameters can therefore be reused with only a slight decrease in accuracy and noise in CNV profiles. We did not experiment with different laboratory protocols; thus, we cannot assess how it may affect the training parameters. The need for re-training for different laboratory processing of the samples and/or sample types makes this approach difficult to test on datasets other than our own since the datasets available usually do not contain enough samples and information to train and test GenomeScreen. The study is based on analyses of 789 training and 106 control samples with both groups of plasma type.

The false-positive rate of GenomeScreen has not been studied in this paper and should be adequately addressed in the future. However, the loss or gain of the (nonmosaic) deviation with a length of at least 100 kb is so substantial that we do not expect to observe any false-positive detections.

One substantial, albeit only technological advantage of the GenomeScreen method, is the involvement of the same laboratories, protocols, chemistries, instruments, and laboratory technicians for both the screening NIPT test and the confirmatory GenomeScreen test. This was not possible in the case of the confirmatory aCGH test due to entirely different protocols, corresponding infrastructure, and chemistry. The ability to use a method and its modifications with the same technical specification for screening as well as diagnostics (subsequent and/or confirmatory) is rarely encountered in laboratory medicine. Therefore, the presented study results fit into the trend of unification of processes on the part of laboratory work as well as bioinformatics and its utilization in different fields of clinical testing.

## 5. Conclusions

In this article, we presented a new method for CNV detection based on low-coverage, whole-genome sequencing—GenomeScreen. We estimated its theoretical sensitivity and conducted a series of in silico tests to estimate it in a semi-real setting. Next, we compared this method directly with a commonly used aCGH method on 48 control samples with known aberrations. The new method detected all of the known aberrations and found even more aberrations mainly in intergenic regions where the studied aCGH delivers poor coverage.

According to the presented results, GenomeScreen is currently able to detect almost all variations longer than 100 kb in mappable regions of the human genome. Moreover, it is cheaper and offers shorter turnaround times in comparison with the studied aCGH method. Thus, in the presented laboratory settings, it represents a favorable replacement for the more conventional aCGH method to detect CNVs longer than 100 kb.

## Figures and Tables

**Figure 1 diagnostics-11-00708-f001:**
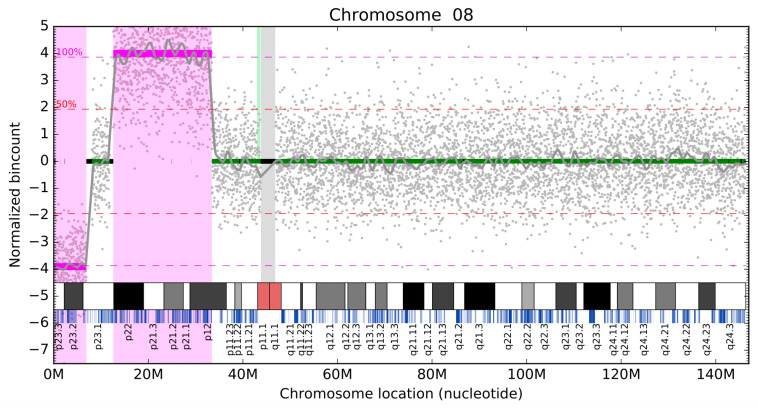
Visualization of the detected deviations on chromosome 8. Chromosome location is on the X-axis. Normalized bin count is on the Y-axis. Green lines represent normal bin count segments (normalized around zero), magenta lines visualize aberrations (one deletion at the start of the chromosome, one duplication on p22–p12). Filtered bins are depicted as black bars on the zero line on the Y-axis. The unmapped region around the centromere is visualized with the grey bar. Grey dots represent the normalized individual bin counts for each bin.

**Figure 2 diagnostics-11-00708-f002:**
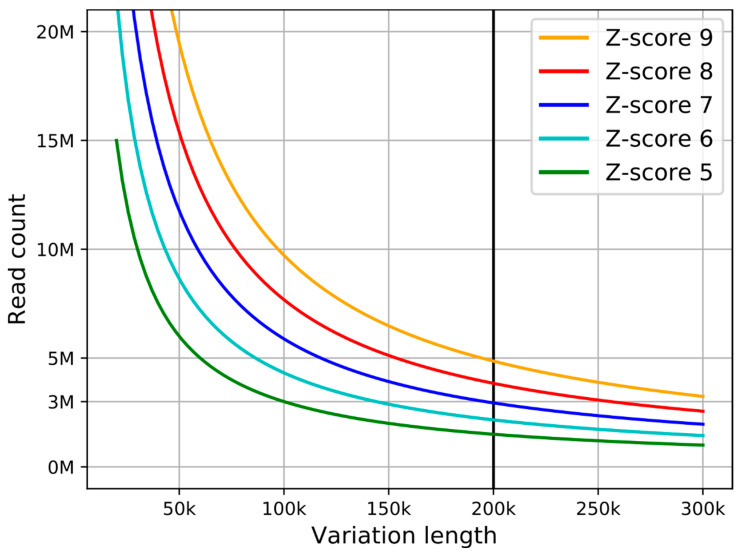
Theoretical minimal read count for successful estimation of copy number variation (CNV) with specified variation length. Different lines represent different Z-score confidence levels.

**Figure 3 diagnostics-11-00708-f003:**
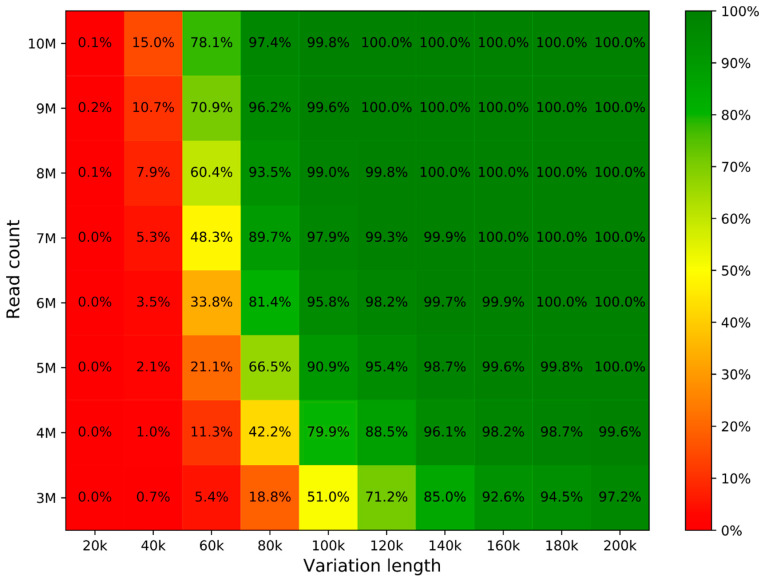
Prediction accuracy computed with in silico analysis based on the length of variation and read count. Each cell number is generated from 8300 simulations (100 randomly generated aberrations; 83 samples).

**Figure 4 diagnostics-11-00708-f004:**
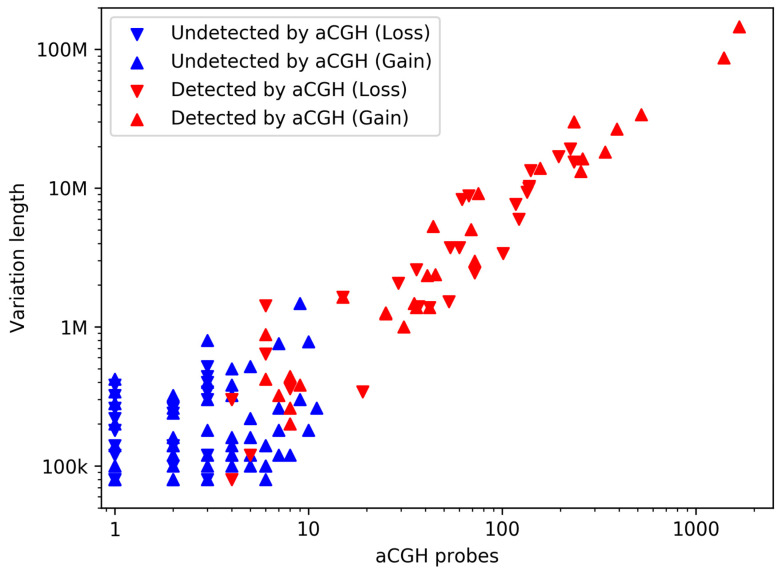
Detection of GenomeScreen (all) and array-based comparative genomic hybridization (aCGH) (red) based on the variation length and number of aCGH probes in the detected interval (by GenomeScreen). Deletions and duplications are visualized by downward and upward triangles, respectively.

## Data Availability

Scripts (Python 3.7) and data are available on the website https://github.com/marcelTBI/GenomeScreen (accessed on 14 April 2021).

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
