# Peer review of "Copy Number Variant Detection with Low-Coverage Whole-Genome Sequencing Represents a Viable Alternative to the Conventional Array-CGH"

_diagnostics, 2021, doi:10.3390/diagnostics11040708_

Round 1

Reviewer 1 Report

The authors have addressed all issues. I have no more question. 

Author Response

The manuscript text was checked by an english proofreader.

Thank you for your review!

Reviewer 2 Report

Two of my major concern was how this method behave while comparing it with other already exist CNV detection methods and can it also identify small CNVs affecting only one or a few small exons with same efficiency? In my view it would have been use-full for the readers to understand the advantage of this method compare to others. However, the authors mentioned in the revised draft that such comparisons are beyond the scope of the manuscript. Thus, I dont have any further comments.

Author Response

We have performed an additional analysis studying the precision of prediction of exact CNV coordinates to respond to the comment about the ability to predict CNVs on the level of exons. 

Added text to Section 3.2: 

Comparison of simulated and reported regions showed that the method can predict the exact simulated region coordinates in 88.2% or coordinates with one-bin difference in 97.7% of cases for 200kb variation length and 10M reads. These numbers slightly drop to 75.3% and 91.7% for 5M reads. The imperfection in predicting coordinates is caused by low coverage and lower mappability of some genomic regions.

In the second paragraph of the Discussion, we discuss the inability of GenomeScreen to work on the level of exons. This would be partially possible by increasing the sequencing coverage and decreasing the bin size but if we would like to decrease the bin size to the median size of exons (100-200bp), then the coverage would need to be greatly increased. Thus it would not be a low-coverage setting anymore.

Unfortunately, the comparison to other CNV tools is beyond the scope of the manuscript as we mentioned.

Furthermore, the text was checked by an english proofreader. 

Thank you for your review!

This manuscript is a resubmission of an earlier submission. The following is a list of the peer review reports and author responses from that submission.

Round 1

Reviewer 1 Report

1.thesis is not unfocused, incomplete
2. unclearly state a purpose
3. low pass WGS and in-silico analysis lack of logicity, and are impractical and irrelevant.

Reviewer 2 Report

Authors build upon their previous work on a CNV detection methods and try to capitalize it by scaling up its potential in comparison to established technology such as aCGH.

The aim is fully legitimate but appears poorly drafted, with poor attention to actual added value. As an example, the first figure displayed is figure 4 (!) and a placemark text "Please add" is still visible in the "Conclusion" section, as if a non definitive version of the manuscript has been submitted.

The scientific soundness of the manuscript is not in doubt and it appears average, while the novelty and actual triggered interest is at least in doubt, or it has not communicated well, since the paper seems to be an upsell of the already published method in the context of aCGH's market.

I suggest a thorough review of the aim in order to clearly differentiate the contribute of this manuscript in comparison to the previous two papers from the same group.

Reviewer 3 Report

Kucharík et at in this manuscript has presented a new NGS based CNV detection method. On the onset, I wanted to let the authors know that in the conclusion section it written “Please Add”. I fail to understand what this mean. The logic behind the method is good; however, authors did not compare their methods with other already exist CNV detection methods (eg. Decon, convading, ExomeDepth etc) and how their method is different from others. Also, authors need to discuss how their method will perform in heterogeneous Mendelian disorders. The authors have mentioned that their tool can identify larger CNVs; but can it also identify small CNVs affecting only one or a few small exons with same efficiency? CNVs are very sensitive to datasets used. Authors should test their methods both in house and another dataset.